# Creation of Mitochondrial Disease Models Using Mitochondrial DNA Editing

**DOI:** 10.3390/biomedicines11020532

**Published:** 2023-02-12

**Authors:** Victoria A. Khotina, Andrey Y. Vinokurov, Mariam Bagheri Ekta, Vasily N. Sukhorukov, Alexander N. Orekhov

**Affiliations:** 1Laboratory of Angiopathology, The Institute of General Pathology and Pathophysiology, 8 Baltiyskaya Street, 125315 Moscow, Russia; 2Laboratory of Cellular and Molecular Pathology of Cardiovascular System, Petrovsky National Research Center of Surgery, Abrikosovsky Lane, 2, 119991 Moscow, Russia; 3Cell Physiology & Pathology Laboratory of R&D Center of Biomedical Photonics, Orel State University, 95 Komsomolskaya Street, 302026 Orel, Russia; 4Laboratory of Medical Genetics, Russian Medical Research Center of Cardiology, Institute of Experimental Cardiology, 15-a 3-rd Cherepkovskaya Street, 121552 Moscow, Russia; 5Institute for Atherosclerosis Research, Osennyaya Street 4-1-207, 121609 Moscow, Russia

**Keywords:** mitochondrial mutations, mitochondrial diseases, cellular model, animal model, gene editing

## Abstract

Mitochondrial diseases are a large class of human hereditary diseases, accompanied by the dysfunction of mitochondria and the disruption of cellular energy synthesis, that affect various tissues and organ systems. Mitochondrial DNA mutation-caused disorders are difficult to study because of the insufficient number of clinical cases and the challenges of creating appropriate models. There are many cellular models of mitochondrial diseases, but their application has a number of limitations. The most proper and promising models of mitochondrial diseases are animal models, which, unfortunately, are quite rare and more difficult to develop. The challenges mainly arise from the structural features of mitochondria, which complicate the genetic editing of mitochondrial DNA. This review is devoted to discussing animal models of human mitochondrial diseases and recently developed approaches used to create them. Furthermore, this review discusses mitochondrial diseases and studies of metabolic disorders caused by the mitochondrial DNA mutations underlying these diseases.

## 1. Introduction

Mitochondrial diseases are hereditary heterogeneous disorders that are based on dysfunctions of the mitochondrial respiratory chain. The presence of mutations in mitochondrial genes can be a reason for these dysfunctions. Mitochondrial DNA (mtDNA) mutations can both appear in somatic cells de novo and be transmitted through the maternal line in generative cells. A unique feature of mtDNA is that for a phenotypic manifestation, it is not necessary to have homoplasmy (a condition in which all mitochondrial DNA molecules carry a mutation) for a specific mutation. In most cases, it is sufficient to achieve a certain degree of heteroplasmy (a condition in which both mutated and wild-type mtDNA molecules are present in the mitochondria) for the manifestation of the disease (mitochondrial threshold effect).

To date, diseases caused by mitochondrial mutations in mtDNA have been insufficiently studied. The main difficulty lies in the insufficient number of patients available to study the molecular basis of the disease [1]. Several cellular models based on the creation of hybrids for the study of mitochondrial diseases have been proposed to circumvent this difficulty. Moreover, it seems promising to use genetic editing techniques (TALEN, CRISPR/Cas9 and others) to create cellular models of mitochondrial disorders. However, cellular models also have certain disadvantages in the study of mitochondrial diseases due to the fact that the cells do not fully exhibit the pathogenic phenotype. Many diseases associated with mitochondrial mutations can manifest themselves only at the tissue or organ level, which makes it possible to fully study them only in animal models. In recent years, animal models for the study of mitochondrial diseases have been gaining popularity. The greatest contribution to the creation of animal models of mitochondrial diseases was the development of a method for the introduction of single-nucleotide substitutions in mtDNA using DddA-derived cytosine base editors (DdCBEs) [2].

In this review, we focus on recent advances in the creation of animal models using mitochondrial genome editing techniques, due to the fact that they are a promising approach to creating a variety of mitochondrial disease models for future studies on the disease pathogenesis and preclinical drug screening. Moreover, this review will discuss mitochondrial diseases and describe well-known mtDNA mutations associated with pathological conditions. Although many valuable models of mitochondrial disease currently exist, they are predominantly based on cells rather than transgenic animals, and few of them mimic the characteristics of pathogenic mtDNA mutations. It is important to note that this review does not elucidate animal models of mitochondrial diseases associated with mutations in nuclear DNA or animal models created using embryo fusion techniques or mutagenesis due to the fact that an excellent and detailed manuscript has already been published on this topic [1].

## 2. Mitochondrial Biology and Functions

It is well known that mitochondria are unique double-membrane-bound organelles that are responsible for a number of functions in cells, such as: (1) the generation of adenosine triphosphate (ATP) and various metabolites through the tricarboxylic acid (TCA) cycle (also known as the citric acid cycle or the Krebs cycle), the electron transport chain (ETC) and oxidative phosphorylation (OXPHOS); (2) apoptosis; (3) calcium homeostasis; (4) nucleotide, cholesterol, glucose, fatty acid, amino acid and heme biosynthesis [3,4,5,6].

Mitochondria consist of the outer mitochondrial membrane (OMM) and the inner mitochondrial membrane (IMM). These phospholipidic membranes divide the mitochondria into the intermembrane space (IMS) and the lumen, which is well known as the “matrix” [7,8]. The OMM and IMM have different lipid compositions, permeability, functions and characteristics. The OMM functions as a diffusion barrier and also mediates signal transduction. The IMM contains anchored ETC complexes, mediates mitochondrial respiration and maintains mitochondrial transmembrane potential (ΔΨ_m_) [9]. The IMM includes the highly packed invaginations in the matrix, mitochondrial cristae and the inner boundary membrane (IBM), which contains various ion channels and transporters.

The OXPHOS system consists of five multimeric protein complexes, also known as respiratory chain complexes (complexes I–IV; CI–IV), and ATP synthase (complex V; CV) embedded in the IMM [9,10]. CI (NADH:ubiquinone oxidoreductase) consists of 45 subunits organized into six modules (N, Q, ND1, ND2, ND4 and ND5) [11]. Seven sub-units of CI form the hydrophobic membrane arm and they are encoded by the mitochondrial genome (*MT-ND1*, *MT-ND2*, *MT-ND3*, *MT-ND4*, *MT-ND4L*, *MT-ND5* and *MT-ND6*), while the rest of the sub-units are encoded by the nuclear genome [11,12]. The N and Q modules form the hydrophilic peripheral matrix arm and include seven “core” subunits (NDUFV1, NDUFV2, NDUFS1, NDUFS2, NDUFS3, NDUFS7 and NDUFS8) which are highly conserved [12,13]. CII (succinate dehydrogenase) has a role not only in the ETC, but also in the TCA [14]. CII consists of four subunits encoded by the nuclear DNA, including the hydrophilic catalytic subunits SDHA/SDH1 and SDHB/SDH2 and the hydrophobic subunits SDHC/SDH3 and SDHD/SDH4 [11,14]. CIII (cytochrome c oxidoreductase) is the key component of the respiratory chain. Cytochrome b (*MT-CYB*) is one of the subunits of CIII that is mitochondria-encoded, while the other CIII subunits are encoded by nuclear DNA [15]. CIV (cytochrome c oxidase) consist of 13 subunits, of which 10 subunits are nuclear-encoded and 3 subunits are encoded by the mtDNA (*MT-CO1*, *MT-CO2* and *MT-CO3*), which are the functional core of the complex [16]. CV (ATP synthase) is the enzyme responsible for the synthesis of the ATP molecule from ADP and phosphate. ATP synthase consists of two domains: the F1 domain, located in the mitochondrial matrix, and the Fo domain, located in the IMM [17,18]. Human CV is composed of 29 proteins, of which only two proteins are encoded by the mtDNA (*MT-ATP6* and *MT-ATP8*) [19].

Mitochondria contain their own mitochondrial DNA (mtDNA), which is located in the mitochondrial matrix [20]. It is a circular 16,569-base pair (bp) double-stranded molecule that is polycistronic and does not contain introns compared to nuclear DNA. mtDNA is found in multiple cellular copies, from 100 to 10,000, that may vary in sequence and quantity among different tissues [21]. mtDNA contains 37 genes encoding 11 messenger RNAs (mRNAs) translated to 13 proteins of the OXPHOS system, 2 ribosomal RNAs (rRNAs; 12S and 16S) and 22 transfer RNAs (tRNAs) [22]. Nevertheless, the human mitochondrial proteome comprises about 1500 proteins [23]. The mitochondrial genetic system requires close interaction between factors encoded by the nuclear and mitochondrial genomes, in contrast to the nuclear genetic system [24]. Thus, mitochondria-encoded protein translation is under dual genetic control and requires the concerted expression of two cellular genomes to fulfill the bioenergetic demands of the cell. Moreover, the translation and transcription of mtDNA require a wide range of cellular protein complexes and transport systems [20,23,25].

The mitochondrial genome has a higher mutation rate about 100–1000 times that of the nuclear genome due to intense redox processes and DNA-damaging free radical formation [26]. mtDNA is not protected by histone proteins, and as a result, defective genes accumulate in it 10–20 times faster than in nuclear DNA [27,28]. Pathological mutations in mtDNA are preserved due to the processes of the fusion and fission of mitochondria, which are essential for the regulation of mitochondrial number, morphology, transport, function and turnover to control the stable state of mitochondria in normal physiological conditions [24]. It is equally important to mention that the elimination of damaged mitochondria is carried out through a process called mitophagy, which also leads to mitochondrial turnover [29,30]. Mitophagy selectively degrades damaged mitochondria and, as a consequence, mediates the cell clearance of mtDNA in normal physiological conditions. In turn, impaired mitophagy contributes to the preservation of mutant mtDNA in cells. As a result, these events may lead to an increase in the heterogeneity of the mtDNA population and the occurrence of more than one type of mtDNA genome in an individual cell or mitochondrion, resulting in a condition termed heteroplasmy. Taken together, all these features lead to the occurrence of mtDNA mutations that have crucial pathological potential for mitochondrial disease development.

## 3. Mitochondrial Diseases

In brief, mitochondrial diseases (MDs) are a heterogeneous group of hereditary diseases caused by mtDNA or nuclear DNA mutations that lead to the structural, molecular-genetic or biochemical dysfunction of mitochondria and the disruption of cellular energy synthesis [31]. Chronic energy deficiency primarily affects the high-energy-demanding tissues and organs, in particular, the central nervous system, heart and skeletal muscles, liver, kidneys and endocrine glands. The clinical symptoms of mitochondrial disease are very different, but the most common manifestations are neurologic, ophthalmologic, audiologic, cardiac, endocrine and renal (Table 1) [32].

The dual genome expression and interactions between the genome products mentioned above contribute to the development of mitochondrial disease in a wide range of organs. The age of mitochondrial disease onset can vary from birth to adulthood, but it is mostly identified in childhood and adolescence [31]. In addition, there are multiple other influences on the age of onset, severity, pattern of organ involvement and progression of the disease.

Mitochondrial diseases can be transmitted by autosomal recessive, autosomal dominant or X-linked inheritance patterns [41]. Mitochondrial diseases originating from mtDNA mutations can be maternally inherited due to the fact that mitochondria and mtDNA are uniparentally transmitted by females [42]. Additionally, it is important to note the role of heteroplasmy in mitochondrial disease development. As mentioned above, each cell contains thousands of copies of mtDNA, and heteroplasmy is characterized by the co-existence of wild-type and mutant mtDNAs within the same cell or mitochondria [21,43]. Heteroplasmy can vary in different tissues and determine the severity of clinical symptoms. The mutation load must exceed a critical threshold of 60–90% to result in a clinical phenotype. The heteroplasmy level is an important property of mitochondrial disease and may vary among patients with the same disease [21,43].

## 4. The Role of mtDNA Mutations in the Development of Mitochondrial Diseases

The clinical phenotypes of various mitochondrial diseases depend on the pathogenic mechanisms, which are determined both by the location of mutations in mtDNA and by the types of cells carrying these mutations (Table 2).

Alterations in mitochondrial function lead to metabolic shifts that increase the production of ATP via alternative pathways, as well as reducing the consumption of ATP for the implementation of energy-consuming processes. An increase in the intensity of glycolysis leads to significant growth in lactate concentration to a critical level for cells [52,73,82]. The chronic toxic effect of lactate was shown on B 82 line-based cytoplasmic hybrids (cybrids) containing a deletion of 4969 bp of mtDNA with heteroplasmy above 40–50%, and may be explained by the disturbance in calcium signaling and mitochondrial biogenesis due to suppression of the expression of PGC1 and Tfam. At the same time, lactate released from cells acts as an intercellular messenger, leading to a decrease in OXPHOS [82]. An increase in the role of glycolysis in ATP production was shown by calculating of the bioenergetic health index ratio of fibroblasts with the mutations m.8993T > G (*MT-ATP6*) m.9185T > C (*MT-ATP6*), m.10158T > C (*MT-ND3*) and m.12706T > C (*MT-ND5*), which are associated with Leigh syndrome, developmental delay, abnormal gait, myopathy, epilepsy, dystonic tetraparesis and severe neonatal lactic acidosis [83]. The observed metabolic changes in the fibroblasts of patients with the m.3243A > G mutation in the *MT-TL* gene can be explained by the activation of the PI3K-AKT-mTORC1 pathway due to increase the NADH/NAD ratio in the cytosol, as well as excessive ROS production, which are associated with mitochondrial dysfunction [52].

Alterations in the metabolism of a number of amino acids have been shown to cause of the development of neurological symptoms in mitochondrial diseases. In particular, patients with MELAS syndrome associated with the m.3243A > G mutation showed an increased concentration of glutamate in the cerebrospinal fluid (CSF) and a decreased concentration in blood plasma, which may be explained by an imbalance in glutamate release and uptake. At the same time, decreased activity of glutamine synthetase, as well as the possibility of the conversion to alpha-ketoglutarate and subsequent metabolism in the TCA for increasing of ATP production, are the main reasons for the high concentration of glutamine in the CSF [47]. Neuronal cybrids, as well as postmortem brain samples of patients with MELAS syndrome with the m.3243A > G mutation, also demonstrated an increase in glutamate concentration, which positively correlated with the level of heteroplasmy and negatively correlated with the activity of CI [48]. It has been shown that metabolic changes are associated with three gene clusters: (1) glutamate and glutamine metabolism; (2) the synthesis, release, reuptake and degradation of gamma-aminobutyric acid; and (3) TCA. High levels of lactate and alanine, as well as low levels of arginine, which are shown in alterations in the gray and white matter of the brain in patients with the m.3243A > G mutation, also indicate metabolic changes (in particular, in NO metabolism) leading to neurological symptoms [84].

In the case of excitable cells (brain or myocardium), mitochondrial dysfunction and energy deprivation caused by mtDNA mutations lead to disturbances in the maintenance of calcium homeostasis. Induced pluripotent stem (iPS) neurons with the m.13513G > A mutation in the *MT-ND5* gene, which leads to respiratory chain deficiency and is associated with Leigh syndrome, have decreased calcium buffering capability, as shown in electrophysiological research [71]. Increased cytoplasmic calcium transient amplitudes were shown in iPS-derived cardiomyocytes harboring the m.3243A > G mutation in the *MT-TL* gene [51].

Mutations of tRNA genes that lead to mitochondrial diseases are characterized by similar mechanisms. Usually, these mutations are localized in regions that are important for the stability of tRNA (experimentally reflected in a decrease in their melting temperatures), tRNA steady-state concentration in mitochondria and aminoacylation activity. This leads to a decrease in the formation of respiratory chain complexes subunits and supercomplexes, a reduction in ΔΨ_m_, cell respiration rate and ATP production, and an increase in ROS production, leading to the development of oxidative stress and, ultimately, apoptosis [44,46,52,54,55,56,58,61,62]. Multiple mtDNA deletions were stimulated for the m.5789T > C mutation [62].

Many pathological mtDNA mutations are associated with alterations in autophagy processes. The m.3460G > A (*MT-ND1*) and m.11778G > A (*MT-ND4*) mutations lead to an increase in the basal level of autophagy (due to modulation of the mTOR/AMPK pathway) and mitophagy (due to elevated accumulation of Parkin in the mitochondria), as well as the absence of a response to autophagy stimulation. These changes correlate with an increase in ΔΨ_m_ and ROS production rates in the mitochondrial matrix [80]. Cybrids with the m.3243A > G mutation with 70% heteroplasmy show the activation of autophagy and mitochondrial biogenesis due to the overexpression of Mitochondrial Nuclear Retrograde Regulator 1 (MNRR1), leading to the compensation of mitochondrial deficiency-related disturbances [49]. Conversely, iPS-derived retinal pigment epithelium cells with the m.3243A > G mutation show a significant decrease in autophagy activity as a result of STAT3 activation and AMPKα suppression. The accumulation of defective mitochondria despite a high level of PINK1 synthesis may be explained by lysosomal dysfunction [50]. Disorders in mitochondrial dynamics, including fission and fusion, in addition to mitophagy, were shown in the results of mitochondrial network state research on fibroblasts with mutations in the *MT-ATP6* (m.8993T > G, m.9185T > C), *MT-ND3* (m.10158T > C) and *MT-ND5* (m.12706T > C) genes [85]. Impaired autophagy and increased apoptosis associated with testosterone influence were also shown for immortalized B lymphocytes with the m.11778G > A mutation in *MT-ND4* from patients with LHON [70]. Defective mitophagy as a pathology factor of the chronification of inflammation was shown for cybrids with the m.12315G > A (*MT-TL2*) and m.14846G > A (*MT-CYB*) mutations [86].

Mutations in the genes of respiratory chain complexes subunits usually lead to a decrease in subunit content in the mitochondria, and, as a consequence, to the dysfunction of complexes or supercomplexes; a decrease in the efficiency of the respiratory chain and ATP production; and elevated ROS generation.

The most common mutations of CI are associated with dysfunctions not only of this CI, but also with combined defects of CI and CIV, CI and CV, and even CI and CII, despite the fact that succinate dehydrogenase is encoded by nuclear DNA [52]. The decrease in the activity of respiratory chain complexes may be the result of reduced content of mitochondrially encoded proteins due to their degeneration, and their dysfunction as a result of molecular conformation defects [63,65,69,70,71,72,73,79,87]. A number of mutations (for example, m.14597A > G in *MT-ND6*) are characterized by an increase in CII activity as compensation for CI dysfunction [73]. This allows us to consider that substrates of CII, for example, succinate, for ATP production increase during both glycolysis and OXPHOS [6]. It should be noted that CI dysfunction can also be associated with mutations in the genes of other mitochondrial proteins, in particular, *MT-ATP6* [88]. One of the pathological manifestations of CI dysfunction is ROS overproduction, including through reverse electron transfer (RET) [9,10]. However, in the case of the mutation m. 14600G > A, which is associated with the development of Leigh syndrome and leads to decreased activity of CI due to its rapid transfer to the deactivated state, ROS production in the presence of the NADH-forming substrates glutamate and malate was almost equal to the control and was reduced in the presence of succinate through RET [89].

Mutations in the genes of ATP synthase subunits that lead to a decrease in mitochondrial function are characterized by alterations in proton transport across the IMM or the degradation of misfolded proteins. In particular, a serine-to-asparagine change in Fo as a result of the m.8969G > A mutation in the *MT-ATP6* gene neutralizes the charge of nearby glutamate, which prevents proton release into the mitochondrial matrix [75]. In the case of the m.9191T > C mutation in the *MT-ATP6* gene, which leads to the replacement of leucine with proline in Fo, the structure of the membrane domain of ATP synthase is disrupted and the efficiency of ATP synthesis decreases by approximately 90% [76].

Mutations in the 12S rRNA and 16S rRNA genes (for example, m.1555A > G and m.2336T > C) lead to alterations in the ribosome structure and corresponding disruption in the synthesis of respiratory chain proteins, a decrease in ∆Ψ_m_ and ATP synthesis, and ROS overproduction [74,90]. Cybrids based on osteosarcoma 143B cells with the mutation m.1555A > G in the *MT-RNR1* gene are characterized by the activation of the AMPK factor and nuclear transcription E2F1, leading to apoptosis of the inner ear cells and deafness [90]. General disturbances in the translation process or defects in the structure and function of individual RC complexes due to an m.1555A > G mutation in the *MT-RNR1* gene lead to the stimulation of mitochondrial biogenesis and accelerated proliferation of Hodgkin’s lymphoma cells [78].

A synergistic effect of several mutations is known. For example, the development of LHON, associated with the mutation m.11778G > A (an arginine-to-histidine change at position 340) in the *MT-ND4* gene, is significantly enhanced by the mutation m. 3866T > C (which leads to the replacement of a highly conserved isoleucine residue at position 187 with threonine in the *MT-ND1* gene). This was confirmed by a significant decrease in ND4 level, the stability and activity of CI and its supercomplexes, the rate of both basal respiration and associated ATP synthesis, respiration, the rate of ATP production, and ∆Ψ_m_ level; moreover, it was associated with increased ROS production and reduced mitophagy [69]. In general, this leads to a significant elevation in the degree of apoptosis. Polycystic ovary syndrome is indicated by a combination of the mutations m.3275C > T, m.4363T > C and m.8343A > G in the *MT-TL*, *MT-TQ* and *MT-TK* genes, respectively, in the study of polymorphonuclear leukocytes [46]. The effect of the m.3394T > C (*MT-ND1*) mutation associated with a CI deficiency, decreased respiratory activity and ATP production, and the development of LHON are significantly enhanced by the m.11778G > A (*MT-ND4*) mutation [62].

## 5. Current Progress in the Development of Cellular and Animal Models of Human Mitochondrial Diseases

The structure and gene organization of mtDNA are highly conserved among mammals, which makes it possible to create various cellular and animal models (Figure 1) of mitochondrial diseases to gain an insight into the pathophysiological mechanisms of the development of these diseases in humans [20,91].

It is worth mentioning that mitochondrial diseases are caused by the presence of mutations not only in mtDNA, but also in nuclear DNA, for example, Leigh syndrome [92]. The most common models of such mitochondrial diseases are animal models with knockout of nuclear genes encoding mitochondrial proteins [93,94,95,96,97]. Such models are clinically relevant and widely used in preclinical studies. Thus, a mouse model of Leigh syndrome was created, obtained by destroying the complex I NADH dehydrogenase-ubiquinone-FeS 4 (*NDUFS4*) gene; this closely replicates the pathological hallmarks observed in patients with this mitochondrial disease, such as growth retardation, hypothermia, ataxia, lethargy and breathing irregularities [93,94,95,97]. In addition, attempts were made to create a model of Leigh syndrome with associated complex IV deficiency in mice and pigs with knockout of the Surfeit Locus Protein 1 (*SURF1*) gene, which is a cytochrome c oxidase assembly factor [96,97].

Currently, the most common approach to creating models of mitochondrial diseases is the use of transmitochondrial techniques, which can be used to create cell lines harboring a pathogenic mitochondrial genome [48,49,86,90,98,99,100,101,102,103,104,105,106,107]. As mentioned above, such cell lines are called cytoplasmic hybrid (cybrid) cell lines (Figure 2). Cybrid cells contain mitochondria with mtDNA mutations from enucleated cells and a nuclear genome of mtDNA depleted cells. Such cell lines have been widely used in studies of the effect of mtDNA on various physiological, pathophysiological and phenotypic biochemical parameters of cells. A number of cybrid cell lines have been created that contain m.3243A > G and m.3271T > C (*MT-TL1*) mutations associated with MELAS, m.14484T > C (*MT-ND6*), m.3460G >A (*MT-ND1*) and m.11778G > A (*MT-ND4*) mutations associated with LHON, the m.8993T > G (*MT-ATP6*) mutation associated with Leigh syndrome and NARP, and m.8344A > G and m.8356T > C (*MT-TK*) mutations associated with MERRF [101,102,103,104,105,106,107]. In addition, cybrid cell lines have been created to investigate the association of mutant mtDNA with Parkinson’s disease and atherosclerosis [86,98,99,100,108].

Moreover, transmitochondrial techniques can be used to create animal models of mitochondrial diseases. Thus, a mouse model with the m.2748A > G mutation (*MT-TL1*) was recently created [109]. The mutation m.2748A > G is an ortholog of the human m.3302A > G associated with polycystic ovary syndrome with insulin resistance, childhood mitochondrial myopathy, encephalomyopathy and MELAS [110,111,112,113]. In this study, enucleated cells harboring mtDNA with the m.2748A > G mutation were fused with female mouse karyotype embryonic stem cells with pharmacologically removed mitochondria, in order to obtain cybrids. Then, these cybrids were transplanted into the oocytes of fertilized C57BL/6J mice [109]. In addition to this model, mouse models with m.6589T > C (*MT-COI*), m.13997G > A (*MT-ND6*), and m.7731G > A (*MT-TK*) mutations were also created earlier using a similar technique [114,115,116]. “Mito-mice” obtained in this way with ≥50% mutant mtDNA heteroplasmy and that exhibit phenotypes similar to mitochondrial diseases can be used to study the molecular basis of pathological phenotypes and mitochondrial dysfunction in various tissues of the body. Despite the advantages of such models, such as high heteroplasmy level, metabolic abnormalities and phenotypes similar to patients, some limitations of “mito-mice” may arise. For example, in “mito-mice” with m.7731G > A and m.6589T > C mutations, hyperglycemia and the formation of ragged-red fibers were not observed, in contrast to patients with mitochondrial diseases, which are often accompanied by the development of diabetes [115,116]. Thus, the precise selection of animal models obtained through the application of transmitochondrial techniques is required for their further use for research purposes.

Other models for studying the mitochondrial diseases mentioned above are iPS cells. Such cell lines are derived from the fibroblasts or peripheral blood mononuclear cells (PBMCs) of patients and have well-characterized naturally occurring mutations in the mitochondrial genome [117]. For example, researchers have created various cell models of MELAS harboring the m.3243A > G mutation (*MT-TL1*), Leigh syndrome harboring the m.9185T > C (*MT-ATP6*) and m.13513G > A (*MT-ND5*) mutations, LHON harboring the m.11778G > A (*MT-ND4*) mutation, and LHON harboring the patient-specific double mutations m.4160T > C (*MT-ND1*) and m.14484T > C (*MT-ND6*) [51,71,118,119,120,121,122,123,124]. Nevertheless, the application of cell lines as relevant models of human mitochondrial diseases has a number of limitations. As mentioned earlier, cell lines can be used to assess the phenotypic and biochemical parameters of a single cell population, while complex intercellular communications occur in the human body. In addition, it is difficult to fully mimic the pathophysiological conditions of the human body underlying the pathology of certain organ systems using cellular models.

According to this, the transgenic manipulation of mtDNA is of great interest for creating mtDNA mutant animal models that can be used to assess the tissue and organ specificity of mutations in mitochondrial diseases, as well as to create therapeutic approaches and undertake preclinical studies of drugs. Some progress has been made in creating mtDNA mutant models using invertebrates such as *Drosophila melanogaster* [125,126]. Thus, lines of *D. melanogaster* containing various homoplasmic mutations in the genes of cytochrome c oxidase subunit I (*MT-COI*) and NADH dehydrogenase subunit 2 (*MT-ND2*) were created using the mitochondria-targeted restriction endonucleases (MitoREs) XhoI (mitoXhoI) and BglII (mitoBglII) [125,126]. These fly models exhibited a wide range of defects, including growth retardation, neurodegeneration, muscular atrophy and reduced life span. These features allow for the use of fly models to study the pathological mechanisms of diseases associated with cytochrome c oxidase deficiency and CI dysfunction, such as Leigh syndrome and MELAS. However, the application of MitoREs has a serious limitation, since they can only be applied to those mtDNA sequences that contain specific restriction sites for bacterial restriction endonucleases [127,128].

In recent decades, significant advances have been made in the field of genetic technologies due to the emergence of engineered programmable nucleases that can be adapted and optimized to study mutations that occur in mtDNA. DNA editing tools such as zinc finger nucleases (ZFNs), transcription activator-like effector nuclease (TALEN) and the clustered regularly interspaced short palindromic repeat (CRISPR)-associated protein 9 (CRISPR/Cas9) system are the most common techniques for genetic manipulation. A number of studies have shown that the development of mitochondria-targeted mitoZFNs, mitoTALENs and mito-CRISPR/Cas9 allows for the site-specific cleavage of mutant mtDNA, followed by heteroplasmy shift, a reduction in mutant mtDNA load and the restoration of wild-type mtDNA levels [129,130,131,132,133,134]. Endonucleases induce the formation of double-strand breaks (DSB) in target sites of mtDNA that contain specific mutations, followed by the elimination of these mtDNAs. This approach can be used to study these mutations and treat genomic pathologies. However, these approaches cannot be fully used to correct homoplasmic mutations or induce mutations in mtDNA de novo, since they are not able to introduce single nucleotide substitutions in mtDNA sequences; this makes them lower-priority tools for the creation of animal and cellular models of mitochondrial diseases. Genome editing technologies such as base editors and prime editors can be used as more precise tools for editing DNA sequences [135,136]. However, such editing techniques require the presence of the CRISPR-Cas protein and guide RNA, which highlights the limitation of mtDNA editing of insufficiently studied pathways of RNA and protein import into the mitochondria [137].

To overcome this challenge, CRISPR-free base editors have recently been developed. An interbacterial deaminase-like toxin (DddA_tox_), derived from the Gram-negative bacterium *Burkholderia cenocepacia*, which catalyzes the deamination of cytidine to uracil in double-stranded DNA (dsDNA) at the TC or TCC sites, has been described [138,139]. RNA-free DddA-derived cytosine base editors (DdCBEs) consist of split-DddA_tox_ halves, mitochondrial targeting sequence (MTS)-linked transcription activator-like effector (TALE) array proteins, and a uracil glycosylase inhibitor (UGI). The split-DddA_tox_ halves are non-toxic and inactive until assembly on the target DNA by adjacently bound programmable dsDNA-binding TALE arrays [139]. DdCBEs catalyze C(G)-to-T(A) conversions in mtDNA with high product purity and specificity (Figure 3). This technique has been found to be applicable to the editing of base pairs in mtDNA, both to shift the heteroplasmy of mutant mtDNA in vitro and to model various single-nucleotide pathogenic substitutions in mitochondrial genes associated with clinical manifestations of mitochondrial diseases.

DdCBEs can be used for the precise creation of cellular and animal models of diseases caused by mtDNA mutations, with typical efficiencies ranging between 5% and 50%, without resulting in a decrease in mtDNA copies in cells. Thus, DdCBE may have greater success in the field of mitochondrial biology and in the potential treatment and therapy of mitochondrial diseases [139].

Nevertheless, this editing tool is limited by the strict TC sequence-context constraint of DddA. Rapid phage-assisted continuous evolution (PACE) and related phage-assisted non-continuous evolution (PANCE) methods were used to increase DdCBE activity in relation to both TC and non-TC targets [140]. This approach allowed for the development of DddA6 and DddA11 variants with conserved mutations enriched during evolution, which mediated a 4.3-fold improvement in the efficiency of mtDNA base editing in TC targets compared to wild-type DddA. In addition, DddA11 has been shown to be highly efficient at editing non-TC targets such as AC and CC when compared to canonical DdCBE. Such improvements in the editing capabilities and enhancement of the efficiency of DdCBE may contribute to the further use of this editing tool for the creation of mitochondrial disease animal models.

Recently, another approach has been developed to overcome this limitation of DdCBE. Thus, transcription-activator-like effector (TALE)-linked deaminases (TALEDs) were created. TALEDs are capable of A-to-G editing in human mitochondria [141]. They consist of custom-designed TALE DNA-binding arrays, a catalytically impaired, full-length DddA variant or split-DddA, and an engineered deoxyadenosine deaminase derived from the *E. coli* TadA protein. TALEDs have been shown to be highly efficient in human cells, with editing frequencies of up to 49% in various mitochondrial genes, and may also be useful for creating cell lines and animal models of mitochondrial diseases.

The selection of an animal species as model organisms to show human pathological conditions should be performed in accordance with the aims and objectives of the planned study. For example, zebrafish (*Danio rerio*) can be selected as an animal model of mitochondrial diseases due to the fact that the human mtDNA sequence is highly conserved. Moreover, both human and *D. rerio* mitochondrial genomes share the same strand-specific nucleotide bias, codon usage and gene order [142]. Thus, five pathogenic human mtDNA mutations were selected to create *D. rerio* models of mitochondrial diseases with mutations such as m.8363G > A (*MT-TK*), m.3733G > A (*MT-ND1*), m.13513G > A (*MT-ND5*), m.12276G > A (*MT-TL1*) and m.3376G > A (*MT-ND1*), which have been associated with MERRF-like syndrome, MELAS, cardiomyopathy, LHON, Leigh syndrome and CPEO [143,144,145,146,147]. It has been demonstrated that the DdCBE editing efficiency of mtDNA can be as high as 88.32% in F0 generation zebrafish. At the same time, mutations introduced by DdCBE into mtDNA can be passed on to 72.5% of the F1 offspring of *D. rerio*, while the mutation load can exceed the pathogenic threshold and be up to 84.33% [143]. In addition, a morphological evaluation of *D. rerio* with introduced mutations revealed the presence of clinical symptoms of mitochondrial diseases. Thus, with aging, individuals with introduced mutations were found to have significant mobility disorders and severe alterations in and fragmentation of the cristae in the mitochondrial matrix.

Recently, an attempt was made to create a model of mitochondrial disease with pathogenetic mtDNA mutations using DdCBE in Sprague Dawley rats [148]. Rats and mice are thought to be phylogenetically closest to humans and other primates. The mutation m.8363G > A (ortholog of rat m.7755G > A) in the mtDNA tRNA^Lys^ gene (*MT-TK*), associated with MELAS, cardiomyopathy and Leigh syndrome, as well as the m.14710G > A mutation (ortholog of rat m.14098G > A) in the tRNA^Glu^ (*MT-TE*) gene, associated with mitochondrial myopathy, were chosen as target pathogenic mutations in this study [144,149]. The editing efficiency in the F0 generation of rats was up to 36.33%, with a mutation load of up to 46.73%, and mtDNAs edited with different efficiencies were found in various tissues of body. In addition, the heritability of the mutations was confirmed in 42.85% of individuals of the F1 generation, with a mutation load of up to 49.15% [148]. Morphological studies and behavioral tests have also shown that rats with mtDNA mutations exhibit clinical symptoms associated with mitochondrial diseases. Thus, it was demonstrated that rats with the m.14710G > A mutation had decreased ATP levels and CI activity in the heart and brain; a dilated cardiomyopathy phenotype; decreased movement distance and average speed; and impaired motor coordination, balance and forelimb grip strength.

However, mice are the most common animal model due to their fecundity and short lifespan. Highly efficient mtDNA editing in C57BL/6J mouse embryos was demonstrated using DdCBE to induce two possible silent mutations: m.12918G > A (ortholog of human m.13513G > A), associated with multiple mitochondrial diseases such as Leigh disease, MELAS syndrome, LHON and LHON/MELAS overlap syndrome in humans, and m.12336C > T, which incorporates a premature stop codon in the *MT-ND5* gene [2,146,150]. Thus, mtDNA editing by DdCBE resulted in the appearance of 25% of C57BL/6J embryos containing the m.12918G > A mutation, with an editing efficiency of up to 23%. The implantation of embryos with the m.12918G > A mutation into surrogate mothers led to the appearance of offspring with the mutant allele, with a frequency of up to 31.6%. In turn, a DdCBE-induced m.12336C > T mutation was observed in 51% of mouse embryos, with an editing frequency of up to 32%. The implantation of embryos with m.12336C > T mutations led to the appearance of offspring with a mutant allele, with a frequency of up to 57% [2]. Moreover, DdCBE-induced mtDNA mutations were found in various tissues of adult mice, thus confirming that mtDNA heteroplasmy was maintained throughout embryonic development and differentiation [2].

Despite all the advantages of DdCBE, there is a high risk of off-target activity due to the presence of a permissive mutant N-terminal domain (NTD) of TALE, which can increase the non-specific binding of TALE arrays, and also due to the fact that DdCBE is a unique type of cytosine deaminase that uses double-stranded DNA as a substrate [139,140,151]. However, the frequency of DdCBE non-specific editing near the target loci of mtDNA remains quite low [139,151].

In order to increase the efficiency of mtDNA editing and avoid the possible off-target editing of base pairs in nuclear DNA, DdCBE was fused with a nuclear export signal (DdCBE-NES) [152]. Thus, a comparison of DdCBE and DdCBE-NES editing efficiency in inducing of the m.12918G > A mutation in the *MT-ND5* gene in C57BL/6J mice showed that the addition of NES to DdCBE increased the efficiency of mtDNA editing by 38.9%, and reduced off-target C-to-T conversions in the nuclear genome from 13.8% to 2%. It is worth noting that both DdCBE-NES and DdCBE contain MTS derived from SOD2 and COX8A and do not have a nuclear localization signal (NLS), which reduces the likelihood of unwanted off-target editing in the nuclear genome [2,152]. It is likely that the use of DdCBE-NES would be the preferred approach for the creation of animal models with mtDNA mutations.

The simultaneous use of DdCBEs and mitoTALENs for the cleavage of unedited mtDNA could increase the frequency of mtDNA editing, which would allow for more precise creation of animal models of mitochondrial diseases [152]. It was demonstrated that the co-injection of mRNA encoding mitoTALEN that targets the wild-type mtDNA sequence, together with mRNA encoding DdCBE or DdCBE-NES, resulted in a 1.7-3-fold increase in editing frequency. Thus, the mouse model was created with the m.12918G > A mutation in the *MT-ND5* gene. The mtDNA-edited mice with the m.12918G > A mutation showed phenotypic and physiological changes with age, such as hunchback, immobility, reduced bodyweight, damage to mitochondria in the kidneys and brown adipose tissue, and asymmetrical hippocampal atrophy resulting in premature death.

In order to enhance the efficiency of base editors, a programmable FusX TALE Base Editor (FusXTBE) system with high editing efficiency in vitro and in vivo has recently been developed [153]. The mitochondria-targeted FusXTBE consist of split-DddA_tox_ halves and UGI fused with a modular FusX that is compatible with the MTS-TALE module. Genes such as *MT-CO1*, *MT-CO3* and *MT-TL1* were used as targets in this study. *D. rerio* was chosen as the most appropriate model organism in this study due to the fact that the amino acid sequences of their *mt-co1*, *mt-co3* and *mt-tl1* proteins displayed >70% similarity to their human orthologues. Mutations in the *MT-CO1* and *MT-CO3* genes are associated with conditions such as Leber optic atrophy, cytochrome c oxidase deficiency, complex IV deficiency, lactic acidosis, encephalopathy, exercise intolerance and myopathy [154,155,156]. In turn, mutations in the *MT-TL1* gene are associated with MELAS disease, which is accompanied by such clinical symptoms as mitochondrial encephalomyopathy, lactic acidosis and stroke-like episodes [157,158]. A new de novo in silico design tool named TALE Writer was developed to predict potential base editing sites in a mitochondrial genome in order to create mutations. Thus, *D. rerio* with the mutations m.7106C > T (*MT-CO1*), m.10215C > T (*MT-CO3*) and m.3744G > A (*MT-TL1*) were created using TALE Writer and FusXTBE. Successful C(G)-to-T(A) conversions at target sites led to the introduction of a stop codon in a protein and tRNA coding genes. FusXTBE has high editing efficiency, as demonstrated by genotyping analysis, which showed that about 70% of the injected embryos showed an editing frequency of up to 90% with the manifestation of mutant mtDNA heteroplasmy and preservation of the mtDNA copy number [153]. In addition, *D. rerio* exhibited a clinical phenotype of patients with pathogenic variations in the *MT-CO1*, *MT-CO3* and *MT-TL1* genes, such as a decreased activity of CI and CIV of up to 72.2% and 73.8%, respectively, and a 2.5-fold increase in lactate levels, indicating mitochondrial dysfunction in individuals with edited mtDNA [153].

Taken together, the results show that precise editing with DdCBE in both rats and mice and *D. rerio* is shown to be a promising tool for creating animal models of human mitochondrial diseases. Additional modifications and improvements in DdCBE, as well as its simultaneous use with mitochondria-targeted engineered programmable nucleases, may increase the efficiency and frequency of mitochondrial genome editing. We can suggest future directions in the field of creating transgenic animals and cell models of mitochondrial diseases and pathological conditions associated with mtDNA mutations due to the powerful methods and approaches at our disposal. One of the possible directions would be to create models of mitochondrial diseases caused by a combination of several mutations in mtDNA, or the simultaneous occurrence of mutations in mtDNA and nuclear DNA [159,160]. Such models could be especially valuable in the development of drugs and therapeutic approaches for patients with rare types of mitochondrial disease. Another direction could be the use of DdCBE as an alternative approach to transmitochondrial techniques for creating cybrids or isolating iPS cells from patients. In this case, the use of DdCBE could enable researchers to overcome the possible limitations that arise during the isolation of cells and tissues from patients with mitochondrial diseases.

Thus, the induction of clinical pathogenic variants of mtDNA mutations in animal models using mtDNA editing techniques, as well as cell lines derived from the cells of patients with mitochondrial diseases (Table 3), will be crucial for understanding the pathogenic mechanisms underlying mitochondrial diseases, for preclinical drug screening, and for the development of therapeutic approaches to human mitochondrial disease treatment.

## 6. Conclusions

Mitochondrial diseases are heterogeneous in severe clinical manifestations and genetic etiology. A growing number of studies over the past decade have identified many variants of mtDNA mutations that lead to the development of mitochondrial dysfunction, followed by the occurrence of defects in cells, tissues and organ systems. However, the development of effective methods for the treatment of mitochondrial diseases is currently a huge problem that is quite difficult to solve without the presence of proper model objects.

To date, some progress has been made in the possibility of creating animal models with pathogenic mtDNA mutations using transmitochondrial techniques. Nevertheless, the development of genetic technologies makes it possible to expand the possibilities of using other methods to create such models. The creation of new transgenic animal models of mtDNA mutation-caused mitochondrial diseases with typical behavioral and anatomical phenotypes of human disorders could provide a greater understanding of the mechanisms of pathogenesis of such diseases. It is likely that the use of mitochondria-targeted editing strategies with DdCBE can help achieve a higher level of control over the creation of transgenic animals.

Thus, the use of new transgenic animals and cell models of mitochondrial diseases, which have experimental advantages and physiological significance, will allow for the development of both preventive measures for patients and the preclinical screening of effective drugs aimed at suppressing the clinical manifestations of such diseases and their accompanying disorders, including diabetes, myopathies, neuropathies and neurodegenerative processes. In addition, animal models of mitochondrial diseases can be actively used to develop approaches to genetic therapy, in which growing interest has appeared in the last few years.

## Figures and Tables

**Figure 1 biomedicines-11-00532-f001:**
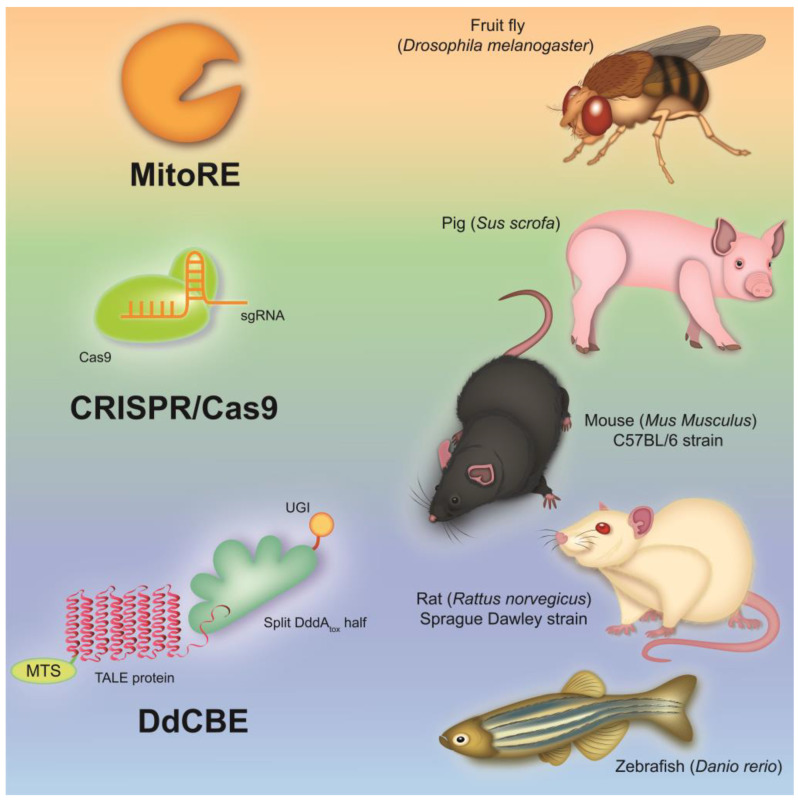
Animal models of mitochondrial disease created through genome editing techniques. There are a number of animal models of mitochondrial diseases caused by defects in nuclear and mtDNA. Thus, to study defects in nuclear-encoded mitochondrial proteins, knockout mice and pigs were created using CRISPR/Cas9. Despite many limitations in their application, models of invertebrate animals, such as the fruit fly, created using mitochondria-targeted restriction endonucleases, are quite common. Nevertheless, the most promising models of mitochondrial diseases at the moment are mice, rats and zebrafish, with mutations in mtDNA introduced by DdCBEs.

**Figure 2 biomedicines-11-00532-f002:**
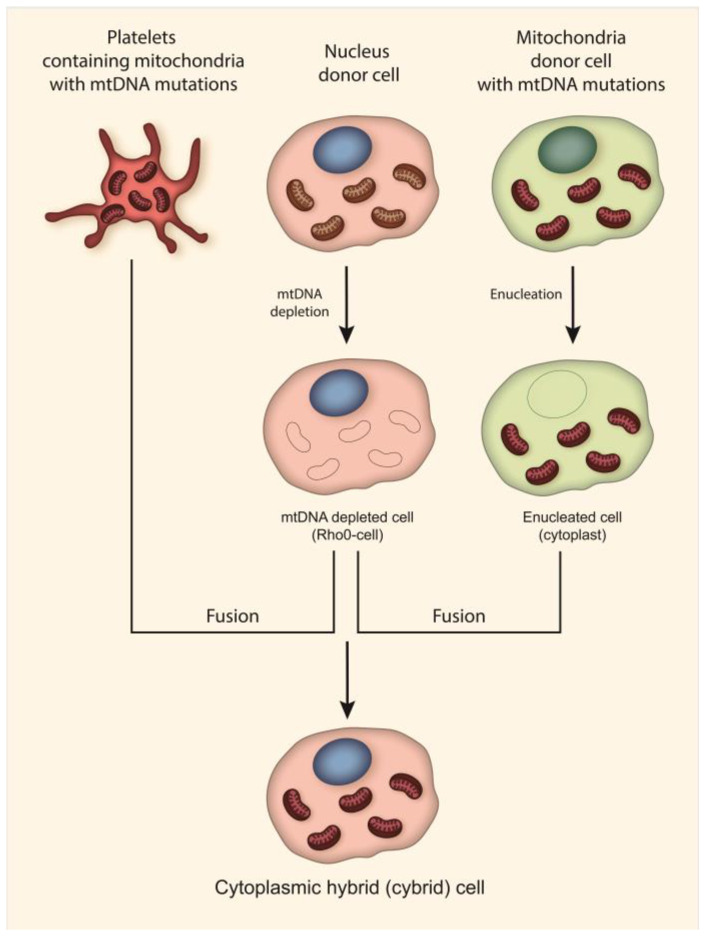
Cybrid cell creation techniques. Cybrids are produced through the fusion of cytoplasm from nucleated cells (Rho0) with enucleated cells (cytoplast) or platelets from patients with mitochondrial diseases. Nucleus donor cells undergo depletion of mtDNA prior to fusion. Thus, cybrids contain mitochondria with mtDNA mutations and a nuclear genome of healthy donors or common cell lines.

**Figure 3 biomedicines-11-00532-f003:**
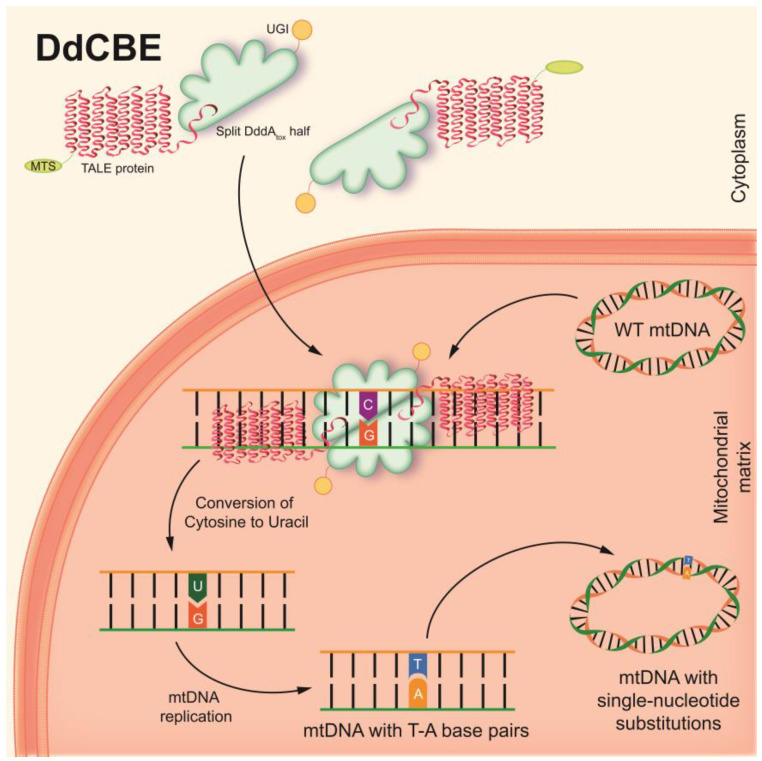
The basic principle of DdCBEs. The two inactive halves of the DdCBE construct import into the mitochondria. After assembly on the target site of the mtDNA by adjacently bound programmable TALE arrays, DdCBE converts cytosine to uracil. UGI protects uracil from the glycosylase until the occurrence of the next round of mtDNA replication. mtDNA replication results in the replacement of uracil by thymine and guanine from the complementary strand by adenine. Thus, DdCBEs catalyze C(G)-to-T(A) conversions in mtDNA without DSB formation with high specificity.

**Table 1 biomedicines-11-00532-t001:** List of the most common genetically defined mitochondrial diseases.

Clinical Syndrome	Clinical Symptoms and Complications	References
Leber Hereditary Optic Neuropathy (LHON)	Acute and painless central vision loss (optic atrophy)	[33]
Leigh syndrome	Bilateral symmetric necrotic lesions in the basal ganglia, brainstem and midbrain, hypotonia, epilepsy, respiratory stress, neurodevelopmental delay, ataxia, lactic acidosis, intellectual decline, movement disorders, headaches, memory loss	[34]
Mitochondrial encephalomyopathy with lactic acidosis and stroke-like episodes (MELAS)	Calcification in basal ganglia, cortical and cerebellar atrophy, stroke-like episodes, seizure, disturbance of consciousness, cognitive impairment, blindness, headache, myopathy, short stature, hemiplegia, cardiomyopathy, gait disturbance, teichopsia, speech disturbance, cerebellar ataxia, diabetes mellitus	[35]
Myoclonus epilepsy associated with ragged-red fibers (MERRF)	Myoclonus, myoclonic seizures, myopathy, sensorineural hearing loss, lipomatosis, dementia, generalized epilepsy, ataxia, ragged-red fibers in the muscle, peripheral neuropathy, renal dysfunction, cardiomyopathy	[36,37]
Maternally inherited diabetes and deafness (MIDD)	Diabetes mellitus, hearing loss	[38]
Neuropathy, ataxia and pigmentary retinopathy (NARP)	Peripheral neuropathy, ataxia, pigmentary retinopathy	[39]
Chronic progressive external ophthalmoplegia (CPEO)	Progressive loss of muscle activity (myopathy), progressive external ophthalmoplegia, ptosis, high-frequency sensorineural hearing loss, progressive dysphagia	[40]

**Table 2 biomedicines-11-00532-t002:** mtDNA mutations associated with mitochondrial diseases and other disorders.

Mutation	Gene	Associated Process	Disease	References
m.616T > C	tRNA^Phe^ (*MT-TF*)	Abolishment of the highly conserved base pair A31-U39 in the anticodon stem reduces levels of ND1, ND4, CO1, CO3, CYTB and ATP8	Isolated chronic kidney disease and hyperuricemia	[44,45]
m.3275C > T	tRNA^Leu^ (*MT-TL1*)	Decreased mtDNA content and mitochondrial membrane potential and 30% reduction in ATP production result in increased ROS production and possible oxidative stress in polymononuclear leukocytes	Polycystic ovary syndrome	[45,46]
m.4363T > C	tRNA^Gln^ (*MT-TQ*)			
m.8343A > G	tRNA^Lys^ (*MT-TK*)			
m.3243A > G m.3271T > C m.3258T > C	tRNA^Leu^ (*MT-TL1*)	Decreased ATP production due to dysfunction of CI cannot support the energy consumption process of glutamate transport and leads to increased glutamate extracellular concentration and excitotoxicity	MELAS	[45,47,48]
m.3243A > G		Mutation-bearing cells have reduced expression of Mitochondrial Nuclear Retrograde Regulator 1 (MNRR1), which, in normal cells, stimulates mitochondrial unfolded protein response, autophagy and mitochondrial biogenesis	MELAS	[45,49]
		iPS-derived retinal pigment epithelium cells with high and low heteroplasmy levels are characterized by a decreased level of basal and CCCP-induced AMPKα activity and lysosomal function and diminished relative autophagy. Mitochondrial dysfunctions, including reduced respiration rate and ATP production, lead to a shift from oxidative phosphorylation to aerobic glycolysis	Age-related macular degeneration in MELAS	[45,50]
		High heteroplasmy mutation level in iPS-derived cardiomyocytes leads to cardiomyopathy associated with reduced respiration rate and ATP production, increased ROS generation in matrix of mitochondria and dysregulation of energy-consuming process of calcium homeostasis	Cardiomyopathy	[45,51]
		Decreased assembly of supercomplexes I2 + III2 + IVn, III2 + IV1 and III2/IV2 and decreased CII level modify glucose metabolism due to constitutive hyperactivation of the PI3K-Akt-mTORC1 pathway	Diabetes, sensorineural deafness and MELAS	[45,52]
		Decreased mitochondrial oxygen consumption, mitochondrial membrane potential and ATP synthesis; increased NADH:NAD ratio	MIDD	[45,53]
m.3256C > T		Dysfunction of the mitochondrial ribosomes due to impossible separation of the newly synthesized proteins	Atherosclerosis	[54]
m.4435A > G	tRNA^Met^ (*MT-TM*)	Decreased levels of ND3, ND4, ND5, ATP6, ATP8, CYTB and CO2	Hypertension	[45,55]
m.5783C *>* T	tRNA^Cys^ (*MT-TC*)	Defective mtDNA replication and decreased mtDNA content lead to instability and reduced activity of the respiratory chain enzymes CI, CIII and CIV and intact supercomplexes	Deafness	[45,56]
m.5789T > C		Reduction in CI and CIV levels, cytochrome c oxidase deficiency in single skeletal muscle fibers and promotion of multiple mtDNA deletions through hybridization between different regions of the mitochondrial genome	NARP	[45,57]
m.5889A > G	tRNA^Leu^ (*MT-TL1*)	Deficient cytochrome c oxidase function, lowered activity of CI and CIII with a tendency towards low activity of CIV and abnormal structure of CV in mitochondria of muscle, but not fibroblasts cells	Childhood-onset severe multi-system disorder characterized by a neurodegenerative course including ataxia and seizures, failure-to-thrive, combined myopathy and neuropathy, and hearing and vision loss	[45,58]
m.8344A > G m.8356T > C m.8361G > A m.8363G > A	tRNA^Lys^ (*MT-TK*)	High COX deficiency and corresponding respiratory insufficiency; decrease in specific tRNA^Lys^ aminoacylation capacity and premature termination of translation at or near each lysine codon	MERRF, CPEO	[37,45,59,60]
m.15927G *>* A	tRNA^Thr^ (*MT-TT*)	Reduced levels of ND1, ND3, ND4, ND5, CO2, CYTB, ATP6 and ATP8	Coronary artery disease	[61]
m.3394T > C	ND1 (*MT-ND1*)	Reduced level, altered assembly and decreased activity of ND1 and subsequent increase in mitochondrial ROS generation and reduction in mitochondrial membrane potential and ATP production	LHON	[45,62]
m.3395A > G		Mitochondrial dysfunction development due to decreased CI quantity because of misfolding protein degradation by ATPase associated with diverse cellular activity of proteases anchored in the inner mitochondrial membrane	Deafness, diabetes and cerebellar syndrome	[45,63]
m.3842G > A		Altered oxidative phosphorylation function leads to ROS-mediated activation of ERK1/2 signaling, increased cell proliferation, migration and invasion, resulting in metastasis promotion	Thyroid cancer	[64]
m.3955G *>* A		Decreased levels of mtDNA-encoded CI subunits (MT-ND4 and MT-ND5) and NDUFB8, mitochondrial respiration activity and mitochondrial membrane potential and increased ROS production in mitochondrial matrix	Leigh syndrome	[65]
m.4160T > C		Significant decrease of not only encoded ND1, but also ND4, ND5, ATP6 and nDNA-encoding subunits SDHB and NDUFB8, leads to basal respiration deficiency due to CI, CII and CIV dysfunctions	LHON plus dystonia	[45,66]
m.5178C > A	ND2 (*MT-ND2*)	Defective function of CI, leading to decreased ATP synthesis and energy deficit	Atherosclerosis and left ventricular hypertrophy	[45,67,68]
m.11778G > A	ND4 (*MT-ND4*)	Destabilization of ND4 structure leads to reduced level of protein, deficient respiration activity, mitochondrial ATP production and mitochondrial membrane potential and increased rate of ROS production in matrix of mitochondria. Cells are characterized by decreased level of mitophagy and promotion of apoptosis. Mutation effect increases under testosterone influence	LHON	[45,69,70]
m.13513G > A	ND5 (*MT-ND5*)	Diminished respiration is associated with decreased mitochondrial mass but not decreased quantity of respiratory chain complexes. Mutant cells are characterized by altered calcium homeostasis due to buffering defect and increased refractory period of neurons	Leigh syndrome	[45,71]
m.14459G > A	ND6 (*MT-ND6*)	Decreased CI activity due to its incorrect assembly in immortalized lymphoblastic cell bearing 39.1% mutated gene	LHON and dystonia overlapping with MELAS episode phenotype	[45,72]
		Energy deprivation due to CI dysfunction leads to atherosclerotic lesion development	Atherosclerosis	[45,54,68]
m.14597A > G		Replacement of Ile with Thr at 26 position leads to decreased CI activity in muscle tissue and skin fibroblasts, ATP production and oxygen consumption rate	Leigh syndrome	[45,73]
m.15059G > A	CYTB (*MT-CYTB*)	Decreased protein length by 244 amino acids leads to CIII dysfunction and to development of atherosclerotic lesions	Atherosclerosis	[45,54,68]
m.652delG	12S rRNA (*MT-RNR1*)	Mitochondrial ribosome dysfunction leads to decreased synthesis of respiratory chain proteins and ATP production, and subsequent energy failure in intimal cells of arteries	Atherosclerosis	[54]
m.1555A > G		Hypermethylation of mitochondrial ribosomes leads to deficient respiration and increased ROS production in matrix of mitochondria that activate AMPK, and subsequently, proapoptotic nuclear transcription factor E2F1 in the stria vascularis and spiral ganglion neurons of the inner ear	Deafness	[45,51]
		Decreased synthesis of respiratory chain proteins due to instability of mitochondrial ribosomes	Atherosclerosis and left ventricular hypertrophy	[45,67,68]
m.2336T > C	16S rRNA (*MT-RNR2*)	Impaired ribosomal assembly due to decreased level of 16S rRNA and ribosomal proteins causes reduced steady-state level of some proteins of respiratory chain (ATP8 and CO2), decreased ATP production and mitochondrial membrane potential level, and elevation of ROS production in matrix of mitochondria	Hypertrophic cardiomyopathy	[45,74]
m.8969G > A	ATP6 (*MT-ATP6*)	Prevention of proton translocation through IMM	Isolated nephropathy followed by a complex clinical presentation with brain and muscle problems	[45,75]
m.9191T > C		90% reduced ATP production prevents ATP synthase subunit a from adopting a stable conformation and makes it prone to proteolytic degradation	Leigh syndrome	[45,76]
m.8597T > C		Oxidative stress due to elevated ROS production and diminished antioxidant status in peripheral mononuclear cells	Type 2 diabetic peripheral neuropathy	[77]
m.8699T > C				
m.8966T > C				
m.10188A > G				
9bp deletion	ND3 (*MT-D3*) COX2 (*MT-CO2*) tRNA^Lys^ (*MT-TK*)			
m.1782G > A	16S rRNA (*MT-RNR2)*	Reduction in steady-state levels of COX1, COX2, COX4, NDUFA9, NDUFV2 and NDUFB8 leads to increased glycolysis rate, but also stimulated mitochondrial biogenesis, and no defects of mitochondrial function and respiratory rate, and as a result, acceleration of cell proliferation and tumor growth	Hodgkin’s lymphoma	[78]
m.8133C > T	COX2 (*MTCO2*)			
m.14512_14513del	ND6 (*MT-D6*)	Dysfunction of CI due to loss of immunoreactive CI subunits (ND1, NDUFV1, NDUFS3 and NDUFB8) in muscle	Exercise intolerance, mild myopathy, deafness and relapsing–remitting neurological presentations	[45,79]
m.3761C > A	ND1 (*MT-D1*)			
m.3460G > A	ND1 (MT-D1)	Excessive autophagy and mitophagy processes in patient-derived cells (fibroblasts, iPS neurons and osteosarcoma cell line-based cybrids) lead to defects in respiration chain and increased ROS production	LHON	[45,80]
m.11778G > A	ND4 (*MT-D4*)			
Large-scale deletion of mtDNA		Decreased mitochondrial oxygen consumption, mitochondrial membrane potential and ATP synthesis; increased ROS production; COX deficiency	CPEO	[53,81]

**Table 3 biomedicines-11-00532-t003:** Animal and cell models of diseases associated with mtDNA mutations.

Model Type	Disease	mtDNA Mutation	Approach	References
Cybrid cells	MELAS	m.3243A > G (*MT-TL1*)	Transmitochondrial technique	[48,49,101,103]
		m.3271T > C (*MT-TL1*)		
		m.14484T > C (*MT-ND6*)		[104]
		m.3460G >A (*MT-ND1*)		
		m.11778G > A (*MT-ND4*)		
	Leigh syndrome NARP	m.8993T > G (*MT-ATP6*)		[105,106]
	MERRF	m.8344A > G (*MT-TK*)		[107]
		m.8356T > C (*MT-TK*)		
	Hodgkin’s lymphoma	m.1555A > G (*MT-RNR1*)		[90]
	Atherosclerosis	m.13513G > A (*MT-ND5*) m.12315G > A (*MT-TL2*) m.3256C > T (*MT-TL1*) m.15059G > A (*MT-CYB*) m.14846G > A (*MT-CYB*) m.1555G > A (*MT-RNR1*)		[86,100,108]
Induced pluripotent stem cells	MELAS	m.3243A > G (*MT-TL1*)	Isolation of fibroblasts or PBMCs from patients	[118,119]
	Leigh syndrome	m.9185T > C (*MT-ATP6*)		[120]
		m.13513G > A (*MT-ND5*)		[71]
	LHON	m.11778G > A (*MT-ND4*)		[121,122,123]
		m.4160T > C (*MT-ND1*)		[124]
		m.14484T > C (*MT-ND6*)		
*Drosophila melanogaster* (fly)	Leigh syndrome MELAS	*mt:CoI^T300I^* and *mt:CoI^R301S^* (*MT-COI*)	MitoRE	[125,126]
		*mt:ND2^Ins1^* and *mt:ND2^Del1^* (*MT-ND2*)		
*Danio rerio* (zebrafish)	MERRF-like syndrome Cardiomyopathy Leigh syndrome	m.8363G > A (*MT-TK*)	DdCBE	[143]
	LHON	m.3733G > A (*MT-ND1*)		
	Leigh syndrome MELAS	m.13513G > A (*MT-ND5*)		
	CPEO	m.12276G > A (*MT-TL1*)		
	LHON	m.3376G > A (*MT-ND1*)		
	Leber optic atrophy Lactic acidosis Encephalopathy Myopathy	zebrafish m.7106C > T (*MT-CO1*)	FusXTBE	[153]
		zebrafish m.10215C > T (*MT-CO3*)		
	MELAS	zebrafish m.3744G > A (*MT-TL1*)		
Mouse C57BL/6J (mito-mice)	Polycystic ovary syn-drome with insulin resistance Childhood mitochondrial myopathy Encephalomyopathy MELAS	m.2748A > G (human m.3302A > G) (*MT-TL1*)	Transmitochondrial technique	[109,114]
		m.6589T > C (*MT-COI*)		
	Diabetes development Lymphoma formation and metastasis	m.13997G > A (*MT-ND6*)		[115]
	MERRF	m.7731G > A (human m.8328G > A) (*MT-TK*)		[116]
	Leigh disease MELAS LHON syndrome LHON/MELAS overlap syndrome	m.12918G > A (human m.13513G > A) (*MT-ND5*)	DdCBE DdCBE-NES Simultaneous use of DdCBE and mitoTALEN	[2,152]
		m.12336C > T (*MT-ND5*)	DdCBE	[2]
Rat (Sprague Dawley)	MELAS Cardiomyopathy Leigh syndrome	m.7755G > A (human m.8363G > A) (*MT-TK*)	DdCBE	[148]
	Mitochondrial myopathy	m.14098G > A (human m.14710G > A) (*MT-TE*)		

## Data Availability

Not applicable.

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
