# Peer review of "Creation of Mitochondrial Disease Models Using Mitochondrial DNA Editing"

_biomedicines, 2023, doi:10.3390/biomedicines11020532_

Round 1

Reviewer 1 Report

The review titled "Creation of mitochondrial disease models by mitochondrial DNA editing" address the state of mitochondrial DNA editing after the discovery of DddA-derived cytosine base editors, and that is the main contribution of this review. Therefore,  the title is proper,  but the summary and extensive parts of the text are redundant and dispersed. For instance, in this paragraph in the summary, "This review discusses mitochondrial diseases and studies of mitochondrial DNA mutation underlying these diseases. Additionally, this review discusses animal models of human mitochondrial diseases and recently developed approaches used to create them" In the first sentence, the authors give one direction to where the review goes, and in the second another direction, now coincident with the title. The segment between lines 169 and 293 discusses the metabolic consequences of mtDNA mutations, which doesn't seem to be the focus of the review. The same occurs between lines 310 and 317, which are also destitute of information citing only two nuclear genes. 

Minor corrections

rephrase lines 98-99

lines 121-132 - the role of mitophagy in heteroplasmy control should be discussed

Table 2 is impressive, but certainly not complete that information has to be cited. 

Author Response

Dear Reviewer,

Coauthors and I very much appreciated the encouraging, critical and constructive comments on this manuscript. Those comments are all valuable and very helpful for revising and improving our manuscript. We strongly believe that the comments and suggestions have increased the scientific value of revised manuscript by many folds. We have taken them fully into account in revision and have made correction which we hope meet with approval. We are submitting the corrected manuscript with the suggestion incorporated the manuscript. Revised portion are highlighted in red in the paper. The manuscript has been revised as per the comments given by the reviewer, and our responses to all the comments are as follows:

Responds to the reviewer’s comments:

Reviewer:

The review titled "Creation of mitochondrial disease models by mitochondrial DNA editing" address the state of mitochondrial DNA editing after the discovery of DddA-derived cytosine base editors, and that is the main contribution of this review. Therefore, the title is proper, but the summary and extensive parts of the text are redundant and dispersed. For instance, in this paragraph in the summary, "This review discusses mitochondrial diseases and studies of mitochondrial DNA mutation underlying these diseases. Additionally, this review discusses animal models of human mitochondrial diseases and recently developed approaches used to create them" In the first sentence, the authors give one direction to where the review goes, and in the second another direction, now coincident with the title. The segment between lines 169 and 293 discusses the metabolic consequences of mtDNA mutations, which doesn't seem to be the focus of the review. The same occurs between lines 310 and 317, which are also destitute of information citing only two nuclear genes.

Response to comment: We are grateful for your review and appreciate your attention to our manuscript. Thank you so much for your comments and remarks. We have rephrased the summary to focus on creation of mitochondrial disease model using different techniques. However, we consider it necessary to discuss the consequences of mtDNA mutations in the context of cellular metabolism so that the readers have an understanding of the pathological mechanisms of the development of mitochondrial diseases caused by the occurrence of mutations in mtDNA. In turn, in the section discussing models of mitochondrial diseases, we consider it equally important to mention information on the most known models of defective and mutated nuclear genes, although this is not the main focus of our review.

Reviewer:

  1. Minor corrections: rephrase lines 98-99

Response to comment: Thank you so much for your comment. We have rephrased the sentence.

  1. Minor corrections: lines 121-132 - the role of mitophagy in heteroplasmy control should be 

    Response to comment: Thank you for pointing this out. We agree with your suggestion and have added more information in revised manuscript (lines 129-133).

    1. Table 2 is impressive, but certainly not complete that information has to be cited.

    Response to comment: Thank you for your comment. We have updated Table 2 with new information and references.

    We tried our best to improve the manuscript and made some changes in the manuscript. We appreciate for Reviewers’ warm work earnestly.

    Once again, thank you very much for all your comments and suggestions.

    Sincerely,

    The authors of manuscript biomedicines-2189618.

Reviewer 2 Report

This review article discussed the mitochondrial diseases in relation to mitochondrial DNA mutation and the animal models for human mitochondrial diseases. Please conduct the concerns below.

1.      In Table 1, title seems better to add “genetic” before “mitochondrial diseases”.

2.      In line 319, the transmitochondrial techniques lacking reference(s).

3.      In Table 3, the title needs to add “in cells”.

4.      Limitations in transmitochondrial techniques during the induction of animal models were not conducted.

5.      How to induce new transgenic animals and cell models? Please give examples in direction even the speculation only.

Author Response

Dear Editor and Reviewers,

Coauthors and I very much appreciated the encouraging, critical and constructive comments on this manuscript by the reviewer. Those comments are all valuable and very helpful for revising and improving our manuscript. We strongly believe that the comments and suggestions have increased the scientific value of revised manuscript by many folds. We have taken them fully into account in revision and have made correction which we hope meet with approval. We are submitting the corrected manuscript with the suggestion incorporated the manuscript. Revised portion are highlighted in yellow in the paper. The manuscript has been revised as per the comments given by the reviewer, and our responses to all the comments are as follows:

Responds to the reviewer’s comments:

Reviewer:

This review article discussed the mitochondrial diseases in relation to mitochondrial DNA mutation and the animal models for human mitochondrial diseases. Please conduct the concerns below.

  1. In Table 1, title seems better to add “genetic” before “mitochondrial diseases”.

Response to comment: Thank you so much for your recommendation. We have added “genetically defined” before “mitochondrial diseases” in Table 1.

  1. In line 319, the transmitochondrial techniques lacking reference(s).

Response to comment: Thank you for pointing this out. We have added lacking references (line 329).

  1. In Table 3, the title needs to add “in cells”.

Response to comment: Thank you so much for your suggestion. We have changed previous Table 3 title to the more appropriate “Animal and cell models of mitochondrial diseases caused by mtDNA mutations”.

  1. Limitations in transmitochondrial techniques during the induction of animal models were not conducted.

Response to comment: Thank you for your valuable recommendation. We have added more information in manuscript (lines 358-365).

  1. How to induce new transgenic animals and cell models? Please give examples in direction even the speculation only.

Response to comment: Thank you so much for your valuable suggestion. We completely agree with your idea. We have added some speculation on these subject to the manuscript (lines 566-577).

We tried our best to improve the manuscript and made some changes in the manuscript. We appreciate for Reviewers’ warm work earnestly.

Once again, thank you very much for all your comments and suggestions.

Sincerely,

The authors of manuscript biomedicines-2189618.
